# Effect of Antibacterial Prophylaxis on Febrile Neutropenic Episodes and Bacterial Bloodstream Infections in Dutch Pediatric Patients with Acute Myeloid Leukemia: A Two-Center Retrospective Study

**DOI:** 10.3390/cancers14133172

**Published:** 2022-06-28

**Authors:** Romy E. Van Weelderen, Kim Klein, Bianca F. Goemans, Wim J. E. Tissing, Tom F. W. Wolfs, Gertjan J. L. Kaspers

**Affiliations:** 1Emma Children’s Hospital, Amsterdam UMC, Vrije Universiteit Amsterdam, 1105 AZ Amsterdam, The Netherlands; k.klein@umcutrecht.nl (K.K.); or gjl.kaspers@amsterdamumc.nl (G.J.L.K.); 2Princess Máxima Center for Pediatric Oncology, 3584 CS Utrecht, The Netherlands; b.f.goemans@prinsesmaximacentrum.nl (B.F.G.); w.j.e.tissing@prinsesmaximacentrum.nl (W.J.E.T.); t.wolfs@umcutrecht.nl (T.F.W.W.); 3Wilhelmina Children’s Hospital, University Medical Center Utrecht, 3584 EA/CX Utrecht, The Netherlands; 4Department of Pediatric Oncology, University Medical Center Groningen, University of Groningen, 9713 GZ Groningen, The Netherlands

**Keywords:** pediatric acute myeloid leukemia, febrile neutropenia, bloodstream infections, viridans group streptococci, Gram-negative rods, antibacterial prophylaxis, teicoplanin, ciprofloxacin

## Abstract

**Simple Summary:**

The intensive chemotherapy that children with acute myeloid leukemia (AML) receive puts them at high risk of infections. Bloodstream infections (BSI) caused by bacteria are common and known for their associated complications but may be prevented by the use of antibacterial agents. Literature on this matter is scarce. We evaluated the effect of different antibacterial prophylaxis regimens on the occurrence of fever and bacterial BSIs in 82 Dutch children with AML. A combination of prophylactic teicoplanin and ciprofloxacin had the best outcomes, resulting in significantly fewer episodes of fever and bacterial BSIs. The combination of teicoplanin and ciprofloxacin was previously suggested by others, but not yet studied. Currently, a randomized trial is ongoing to address and validate the efficacy of teicoplanin prophylaxis in pediatric AML.

**Abstract:**

Bloodstream infections (BSIs), especially those caused by Gram-negative rods (GNR) and viridans group streptococci (VGS), are common and potentially life-threatening complications of pediatric acute myeloid leukemia (AML) treatment. Limited literature is available on prophylactic regimens. We retrospectively evaluated the effect of different antibacterial prophylaxis regimens on the incidence of febrile neutropenic (FN) episodes and bacterial BSIs. Medical records of children (0–18 years) diagnosed with de novo AML and treated at two Dutch centers from May 1998 to March 2021 were studied. Data were analyzed per chemotherapy course and consecutive neutropenic period. A total of 82 patients had 316 evaluable courses: 92 were given with single-agent ciprofloxacin, 138 with penicillin plus ciprofloxacin, and 51 with teicoplanin plus ciprofloxacin. The remaining 35 courses with various other prophylaxis regimens were not statistically compared. During courses with teicoplanin plus ciprofloxacin, significantly fewer FN episodes (43 vs. 90% and 75%; *p* < 0.0001) and bacterial BSIs (4 vs. 63% and 33%; *p* < 0.0001) occurred than with single-agent ciprofloxacin and penicillin plus ciprofloxacin, respectively. GNR and VGS BSIs did not occur with teicoplanin plus ciprofloxacin and no bacterial BSI-related pediatric intensive care unit (PICU) admissions were required, whereas, with single-agent ciprofloxacin and penicillin plus ciprofloxacin, GNR BSIs occurred in 8 and 1% (*p* = 0.004), VGS BSIs in 24 and 14% (*p* = 0.0005), and BSI-related PICU admissions were required in 8 and 2% of the courses (*p* = 0.029), respectively. Teicoplanin plus ciprofloxacin as antibacterial prophylaxis is associated with a lower incidence of FN episodes and bacterial BSIs. This may be a good prophylactic regimen for pediatric AML patients during treatment.

## 1. Introduction

Although the current, highly intensive, chemotherapeutic regimens used in pediatric acute myeloid leukemia (AML) have contributed to immense improvement in overall survival rates [1,2,3], the associated serious treatment-related toxicities remain an important concern [4,5]. In particular, the chemotherapy-induced severe and prolonged neutropenia predisposes patients to bacterial and fungal infections, which may cause life-threatening complications throughout treatment and significant treatment delay [6,7,8,9].

Bloodstream infections (BSIs) caused by Gram-negative rods (GNR) and the Gram-positive viridans group streptococci (VGS) are common and contribute largely to the treatment-related mortality of children with AML [9,10,11]. Although the bacterial infection-related mortality rates have decreased over the past years [7,12,13,14,15,16,17,18] as a result of prompt initiation of empiric systemic antibiotics, the use of antibacterial prophylaxis and very intensive supportive care otherwise, there is no international consensus on the optimal infection prophylaxis regimen.

The few available studies evaluating the efficacy of antibacterial prophylaxis regimens in pediatric AML are mainly single-centered with small sample sizes and are often limited by their retrospective design and the comparison of only two regimens [13,14,15,16,17,18,19,20,21,22]. The most effective prophylactic antibacterial agents seem those with Gram-positive coverage, such as vancomycin or teicoplanin [13,22], or a combination of an agent with Gram-positive coverage and an agent with Gram-negative coverage [15,17,23]. In a randomized clinical trial, the effect of the fluoroquinolone levofloxacin, which has activity against both Gram-positive and Gram-negative bacteria, has been compared with no prophylaxis [24]. Although levofloxacin was significantly associated with a decrease in bacterial BSI incidence per pediatric AML patient, GNR and VGS BSIs were not eliminated [24]. Recently, a clinical practice guideline weakly recommended levofloxacin for systemic antibacterial prophylaxis in pediatric AML with the note that the institution’s antimicrobial resistance patterns and the potential fluoroquinolone-related adverse effects need to be taken into account [25].

In the previous two decades, different antibacterial prophylaxis regimens have been used in Dutch pediatric AML patients. In this two-center retrospective study, we aimed to evaluate the effect of these regimens on the incidence of episodes of febrile neutropenia (FN) and culture-proven bacterial BSIs.

## 2. Materials and Methods

### 2.1. Patients

Patients were eligible if they were between 0–18 years of age at diagnosis, newly diagnosed with AML at the VU University Medical Center (VUmc) in Amsterdam, The Netherlands between May 1998 and June 2018, or at the Princess Máxima Center (Utrecht, The Netherlands) for Pediatric Oncology in Utrecht, The Netherlands between June 2018 and March 2021, and treated according to one of the Dutch Childhood Oncology Group (DCOG) (Leyweg, Netherlands) treatment protocols in use at the time of diagnosis. Patients who were diagnosed with acute promyelocytic leukemia, myeloid leukemia of Down syndrome, secondary AML, died within the first week of treatment, or did not have an evaluable chemotherapy course (see definitions) were excluded. The informed consent forms of the applied protocols included written informed consent of the parents or legal guardians and/or patient for data collection for scientific purposes. This study was approved by the Institutional Review Board and Ethics committee of both centers.

### 2.2. Data Collection

The DCOG (Leyweg, Netherlands) registry provided a list of eligible patients. The scanned medical records of patients diagnosed before 2016 and the electronic medical records of all other eligible patients were studied. A case report form was filled out per patient including patient-specific, AML biology-related, and treatment-related features. Data on FN episodes, blood cultures, isolated microorganisms, antimicrobial susceptibility, bacterial BSI-related pediatric intensive care unit (PICU) admissions, and the use of empiric systemic and prophylactic antibiotics were collected per chemotherapy course and the consecutive neutropenic period. Data were collected from the first day of the first chemotherapy course until protocol completion with absolute neutrophil count (ANC) recovery ≥0.5 × 10^9^/L post nadir, the start of treatment for refractory disease or relapse, start conditioning prior to stem cell transplantation, or death, whichever occurred first.

### 2.3. Pediatric AML Treatment

Within the study period, patients were treated with four consecutive DCOG treatment protocols: ANLL-97/AML-12 [26] (May 1998–January 2007), AML-15 [27] (January 2007–June 2010), DB-AML-01 [2] (June 2010–January 2014), and NOPHO-DBH AML 2012 [28] (ClinicalTrials.gov identifier NCT01828489) (January 2014–March 2021). Appendix A shows protocol details.

### 2.4. Antimicrobial Prophylaxis

During the study period, different antibacterial prophylaxis regimens have been used. In the early years (May 1998–April 2006), prophylaxis included either single-agent penicillin, which was administered in case of penicillin-sensitive VGS strains in surveillance throat cultures or no prophylaxis at all if the VGS strains were either absent or intermediate-sensitive/resistant to penicillin. In May 2006, ciprofloxacin prophylaxis was introduced. From May 2006 to May 2010, prophylaxis included either penicillin plus ciprofloxacin or single-agent ciprofloxacin. Between June 2010 and June 2018, teicoplanin (three loading doses of 6–12 mg/kg every 12 h, followed by a daily maintenance dose of 6–12 mg/kg in one dose based on the patients’ age) was added to ciprofloxacin in patients with intermediate-sensitive/resistant to penicillin VGS strains in the surveillance throat cultures. Patients who were not admitted to the hospital received teicoplanin at the outpatient clinic or via domiciliary care. Two of the 19 patients who received teicoplanin developed a maculopapular rash indicated as a hypersensitivity reaction to teicoplanin. In these patients, a macrolide (clarithromycin or azithromycin) was combined with ciprofloxacin. Until June 2018, the Gram-positive prophylaxis with either penicillin, teicoplanin, or a macrolide was started after myelosuppressive treatment, or when the ANC was ≤0.5 × 10^9^/L and continued until ANC recovery to ≥0.5 × 10^9^/L post nadir, or the start of the next chemotherapy course. From June 2018 onwards, oncology care was centralized in the Princess Máxima Center (Utrecht, The Netherlands) for Pediatric Oncology, and penicillin plus ciprofloxacin became the standard prophylactic regimen. Penicillin was started halfway through the first chemotherapy course or at an ANC ≤0.5 × 10^9^/L, whichever occurred first, and at the first or last day of the consecutive courses and continued until an ANC ≥0.5 × 10^9^/L post nadir, or the start of the next chemotherapy course. A macrolide (clindamycin) was given instead of penicillin if the patient had experienced a BSI with a penicillin-resistant VGS strain. Because of a consensus on a lack of evidence of the prophylactic effect of penicillin, the standard prophylactic regimen changed to single-agent ciprofloxacin from June 2020 onwards. In some patients, the antibacterial prophylaxis regimen was switched throughout their course of AML treatment or during the neutropenic period following a chemotherapy course. As ciprofloxacin, teicoplanin, and macrolides do not have an approved indication to prevent bloodstream infections in children in Europe, the administration of these antibiotics as prophylaxis was off-label.

According to protocol, all patients received *Pneumocystis jiroveci* prophylaxis with co-trimoxazole and antifungal prophylaxis with itraconazole (suspension) or other triazoles; in some cases (liposomal) amphotericin B, or an echinocandin (micafungin or caspofungin) were prescribed. Granulocyte colony-stimulating factor was not routinely used.

Patients were discharged after completion of the chemotherapy course if they were in good clinical condition. In case of FN or other toxicities needing admission, patients were readmitted. Blood cultures were routinely collected via the central venous line and empiric systemic antibiotics were started according to local guidelines. Prophylactic antibiotics were mostly discontinued and resumed if blood cultures were negative for 72 h and the patient was afebrile for at least 24 h.

### 2.5. Definitions

Fever was defined as a single body temperature ≥38.5 °C or two times ≥38.0 °C one hour apart. FN was defined as fever with an ANC ≤0.5 × 10^9^/L. BSI was defined as a positive blood culture and signs of infection. Positive blood cultures that were considered contaminants by the hospital’s microbiologist were excluded. The isolates of polymicrobial BSIs (two or more microorganisms isolated from a single blood culture) were analyzed separately, but the BSI was counted as a single event. If a patient experienced multiple BSIs during a chemotherapy course, these BSIs were considered separate infections unless the same microorganism was cultured ≤7 days apart. Chemotherapy courses were not considered evaluable if empiric systemic antibiotics were started along with chemotherapy and continued the whole aplastic period or until the patient died, the patient died before the chemotherapy course was completed, or if the chemotherapy course was agent-adjusted resulting in a treatment-related protocol violation. PICU admissions were considered bacterial BSI-related if the reason for admission in the medical record stated respiratory and/or circulatory insufficiency due to bacterial sepsis.

### 2.6. Statistical Analysis

Data were described as absolute numbers and as percentages. Medians and interquartile ranges (IQR) were used to report non-normally distributed data. If the antibacterial prophylaxis regimen switched during the neutropenic period following a chemotherapy course, both prophylaxis regimens were considered evaluable and included in the analysis. Pearson’s χ^2^ or Fisher’s exact tests were used to test differences in proportions of FN episodes, bacterial BSIs, and bacterial BSI-related PICU admissions between chemotherapy courses given with single-agent ciprofloxacin, penicillin plus ciprofloxacin, and teicoplanin plus ciprofloxacin. Data on (infection-related) deaths were collected up to April 2022. Two-sided *p*-values < 0.05 were considered statistically significant. IBM SPSS version 26 was used for data analysis.

## 3. Results

A total of 85 patients were eligible, of whom three were excluded because of death before the onset of treatment (*n* = 2) and not having an evaluable chemotherapy course (*n* = 1). Of the 82 included patients, 45 (54.9%) were male, the median age was 9 (IQR 3–13) years, and median white blood cell count at diagnosis was 14.3 (IQR 5.4–49.6) ×10^9^/L (Table 1).

Included patients received 322 chemotherapy courses, of which 19 (5.9%) were not evaluable and excluded from analysis (due to the administration of empiric systemic antibiotics along with chemotherapy during the whole aplastic period *n* = 16 or until the patient died *n* = 1, death before completion of the chemotherapy course *n* = 1, or a treatment-related protocol violation *n* = 1). Hence, 303 chemotherapy courses were evaluated: 69 (22.8%) first induction (induction I), 76 (25.1%) second induction (induction II), 65 (21.5%) first consolidation (consolidation I), 54 (17.8%) second consolidation (consolidation II), and 39 (12.9%) third consolidation (consolidation III) courses.

In 13 (4.3%) of the 303 evaluable courses, the antibacterial prophylaxis regimen switched during the neutropenic period. Altogether, 316 courses were evaluated for the effect of antibacterial prophylaxis, of which 92 (29.1%) were given with single-agent ciprofloxacin, 138 (43.7%) with penicillin plus ciprofloxacin, and 51 (16.1%) with teicoplanin plus ciprofloxacin. Additionally, 10 (3.2%) courses were given with no prophylaxis at all, 12 (3.8%) with single-agent penicillin, and 13 (4.1%) with a macrolide plus ciprofloxacin (Table 2). Because of the small number of courses with the latter three regimens, these were only described and not statistically compared.

FN episodes occurred significantly less often with teicoplanin plus ciprofloxacin (43.1%, 22/51 courses) than with single-agent ciprofloxacin (90.2%, 83/92 courses) and penicillin plus ciprofloxacin (75.4%, 104/138 courses) (*p* < 0.0001). The incidence of bacterial BSIs was significantly lower with teicoplanin plus ciprofloxacin (3.9%, 2/51 courses) than with single-agent ciprofloxacin (63%, 58/92 courses) and penicillin plus ciprofloxacin (33.3%, 46/138 courses) (*p* < 0.0001). VGS BSIs did not occur during any of the courses with teicoplanin plus ciprofloxacin (0/51), whereas the incidence of VGS BSIs was 23.9% (22/92 courses) with single-agent ciprofloxacin and 13.8% (19/138 courses) with penicillin plus ciprofloxacin (*p* = 0.0005). Similarly, GNR BSIs did not occur during any of the courses with teicoplanin plus ciprofloxacin (0/51) compared with 7.6% (7/92 courses) of the courses with single-agent ciprofloxacin and 0.7% (1/138 courses) of the courses with penicillin plus ciprofloxacin (*p* = 0.004). Similar results were obtained if the 13 courses during which the antibacterial prophylaxis regimen switched during the neutropenic period were excluded (Appendix A).

In 106 (33.5%) of the 316 evaluable courses, empiric systemic antibiotics were initiated along with chemotherapy but discontinued prior to the onset of prophylactic antibiotics. If these courses were excluded from analyses, similar results were obtained (Appendix A).

In total, 135 BSIs were registered in 62 (75.6%) patients and within 130 (42.9%) courses. Five courses included two BSIs. BSIs occurred most frequently during induction II (*n* = 42, 31.1%), followed by 30 (22.2%) BSIs during both induction I and consolidation I, 24 (17.8%) during consolidation II, and 9 (6.7%) during consolidation III. Of the 135 BSIs, 114 (84.4%) were monomicrobial and 21 (15.6%) polymicrobial, resulting in 161 isolated microorganisms. Table 3 shows the microorganisms cultured in these BSIs. Gram-positive bacteria were most commonly detected (85.7%) with coagulase-negative staphylococci (CoNS) (44.1%) and VGS (29.8%) as predominant pathogens, followed by Gram-negative bacteria (9.9%), and fungi/yeasts (4.3%). Over the course of treatment, 34 (41.5%) patients experienced a VGS BSI. Eight (23.5%) patients had two VGS BSIs during two different courses and three (8.8%) patients had three VGS BSIs during three different courses. Multiple VGS BSIs during the same course were not experienced by any of the patients.

The incidence of FN episodes was similar after courses with low-dose (LD) (63.2%, 12/19 courses) and high-dose (HD)-cytarabine (78.9%, 224/284 courses) (*p* = 0.149). Bacterial BSIs occurred significantly more often after HD-cytarabine (43.0%, 122/284 courses) than after LD-cytarabine courses (15.8%, 3/19 courses) (*p* = 0.028) (Table 4). Among LD-cytarabine courses, there were no significant differences in the incidences of FN episodes and bacterial BSIs between single-agent ciprofloxacin, penicillin plus ciprofloxacin, and teicoplanin plus ciprofloxacin. Among HD-cytarabine courses, teicoplanin plus ciprofloxacin was associated with significantly fewer FN episodes (39.5%, 17/43 courses vs. 89.8%, 79/88 courses vs. 78.3%, 101/129 courses; *p* < 0.0001) and significantly less bacterial BSIs (4.7%, 2/43 courses vs. 64.8%, 57/88 courses and 34.1%, 44/129 courses; *p* < 0.0001) compared with single-agent ciprofloxacin and penicillin plus ciprofloxacin, respectively.

Table 5 shows the penicillin susceptibility of the VGS isolates and per antibacterial prophylaxis regimen. Of 16/21 VGS isolates cultured during courses with either single-agent penicillin or penicillin plus ciprofloxacin prophylaxis, the penicillin-susceptibility was known, 10 (62.5%) of which were intermediate-sensitive/resistant. Moreover, in six (28.6%) cases, a VGS BSI occurred despite penicillin prophylaxis and a penicillin-sensitive VGS strain.

The 16 Gram-negative BSIs occurred under single-agent ciprofloxacin (*n* = 7), single-agent penicillin (*n* = 5), no prophylaxis (*n* = 3), and penicillin plus ciprofloxacin prophylaxis (*n* = 1). For 6/8 Gram-negative strains that were cultured under either single-agent ciprofloxacin or penicillin plus ciprofloxacin prophylaxis, the ciprofloxacin susceptibility was tested, three (50%) of which were ciprofloxacin-resistant. The ciprofloxacin susceptibility was not tested for the Gram-negative strains that were cultured under no prophylaxis and single-agent penicillin.

Musculoskeletal toxicities were not reported in any of the patients who received ciprofloxacin prophylaxis.

In total, there were 15 bacterial BSI-related PICU admissions, of which eight (53.3%) were due to infectious complications caused by VGS. Overall, seven (20.6%) of 34 patients with a VGS BSI had to be treated at the PICU. Bacterial BSI-related PICU admissions were not required during any of the courses with teicoplanin plus ciprofloxacin (0/51) compared with 7.6% (7/92) of the courses with single-agent ciprofloxacin and 2.2% (3/138) of the courses with penicillin plus ciprofloxacin (*p* = 0.029) (Table 2). At the time of analysis, 19 (23.2%) out of the 82 included patients had died: six (31.6%) deaths were BSI-related (*n* = 4 during initial treatment and *n* = 2 during relapse treatment). Two (28.6%) of these BSI-related deaths involved VGS and involved patients who received single-agent penicillin (*n* = 1) and penicillin plus ciprofloxacin (*n* = 1). The isolated VGS strains were sensitive and intermediate-sensitive to penicillin. Prior to the introduction of ciprofloxacin prophylaxis, there were three BSI-related deaths caused by Gram-negative bacteria. One BSI-related death was caused by *Candida Krusei* accompanied by endocarditis.

## 4. Discussion

Our study shows that a combination of teicoplanin and ciprofloxacin as antibacterial prophylaxis is associated with a significantly reduced incidence of FN episodes and bacterial BSIs in Dutch pediatric AML patients. Moreover, no bacterial BSI-related PICU admissions or -deaths were reported with this regimen.

Over the past decades, in pursuit of improving survival and decreasing infection-related morbidity and mortality, various antibacterial prophylaxis regimens have been applied during pediatric AML treatment. However, these regimens are not routinely used because of concerns regarding the emergence of antibiotic resistance, locally different resistance patterns, differences in BSI incidences between countries and even centers, availability of specific antibacterial agents, agent-associated toxicity, possible increase in fungal infections, costs, and lack of evidence from pediatric studies in general.

In adult cancer patients who are at high risk of severe and prolonged neutropenia, prophylaxis with ciprofloxacin or levofloxacin is recommended, with a preference for levofloxacin if the risk of mucositis-related VGS BSIs is high [29]. The addition of an agent with Gram-positive coverage is not usually recommended [29]. The few retrospective pediatric AML studies that compared the effect of ciprofloxacin prophylaxis with no prophylaxis [16,18] showed fewer GNR BSIs, but more Gram-positive BSIs with ciprofloxacin use. One study reported the use of ciprofloxacin prophylaxis as an independent adverse prognostic factor for the occurrence of VGS BSIs [16]. With regard to levofloxacin, one randomized trial showed a significantly lower bacterial BSI incidence per pediatric AML patient with the use of levofloxacin (23%) compared with no prophylaxis (40%) [24]. However, among acute leukemia patients in the levofloxacin group, VGS and GNR were detected in 54 and 21% of positive blood cultures, respectively, compared with 49 and 43% in the no prophylaxis group. These infections were thus not eliminated [24]. Potential, mainly reversible, fluoroquinolone-related side effects are musculoskeletal toxicities, which limit the fluoroquinolone use in children. In our study, musculoskeletal toxicities were not reported in any patient receiving ciprofloxacin prophylaxis, which was in agreement with two other studies on the efficacy of ciprofloxacin prophylaxis in childhood AML [16,18]. A few other retrospective studies among pediatric AML patients have shown that cefepime monotherapy [14] or vancomycin combined with an agent with Gram-negative coverage are highly effective in reducing bacterial BSIs [14,15,17,20,23]. However, vancomycin is associated with dosing difficulties and infusion-related side effects and requires administration (at least) every 12 h with therapeutic drug monitoring. In this regard, teicoplanin, a glycopeptide with a spectrum of activity similar to that of vancomycin, seems more attractive as it can be administered once daily and is associated with fewer side effects. One study analyzed the efficacy of the prophylactic use of teicoplanin on alternate days (85/98 courses) and vancomycin monotherapy (13/98 courses) as being one regimen (teicoplanin/vancomycin) [13]. It showed that with this regimen FN episodes were significantly reduced and VGS BSIs were completely eradicated in the pediatric AML population compared with no prophylaxis [13]. However, due to the occurrence of GNR BSIs, there was no net difference in the incidence of bacterial BSIs between teicoplanin/vancomycin and no prophylaxis [13]. Given the available studies and our results, a regimen including an agent with Gram-positive coverage and an agent with Gram-negative coverage seems most justifiable, as no single agent seems sufficiently effective in eliminating both BSIs caused by GNR and VGS.

Our results support the use of such a two-agent regimen. The combination of penicillin and ciprofloxacin was significantly associated with a lower incidence of FN episodes and bacterial BSIs compared with single-agent ciprofloxacin. Conventionally, VGS have been presumed to be susceptible to penicillin [30]. However, penicillin-resistance of VGS isolates is emerging and resistance rates range from 20–100% [13,15,31]. In our study, VGS BSIs occurred in 14% of the courses given with penicillin plus ciprofloxacin. This can be explained by the fact that 64% of the VGS strains tested in these courses were intermediate-sensitive/resistant to penicillin and possibly partly by the non-adherence to oral penicillin, as it is not well-tolerated. Overall, about 50% of VGS strains in our study were intermediate-sensitive/resistant to penicillin. Based on these findings, penicillin may thus not be the most attractive agent with Gram-positive coverage to add to ciprofloxacin.

Compared with single-agent ciprofloxacin and penicillin plus ciprofloxacin, teicoplanin plus ciprofloxacin was significantly associated with fewer FN episodes and bacterial BSIs. Notably, GNR and VGS BSIs were completely eliminated with the use of teicoplanin plus ciprofloxacin and only two bacterial BSIs occurred caused by CoNS. These results seem particularly relevant as Gram-positive pathogens, mainly CoNS and VGS, are currently the leading cause of BSIs. This is probably a result of intensive chemotherapeutic regimens with HD-cytarabine, the introduction of fluoroquinolone prophylaxis, the use of central venous lines, and chemotherapy-induced imbalance in the anaerobic and aerobic gut and oral bacteria, and mucosal barrier injury [32,33].

Previously, HD-cytarabine has been considered a risk factor for the occurrence of VGS BSIs [34]. This has been supported by some studies [7,13], but not by others [8,31]. In our study, all VGS BSIs occurred during courses with HD-cytarabine (48/284), but this was not significantly different from the courses with LD-cytarabine (0/19) (*p* = 0.052), most likely because of the small number of LD-cytarabine courses. These varying findings may be explained by the use of different antibacterial prophylaxis regimens in studies, different definitions for HD-cytarabine, and different treatment protocols. The favorable effect of teicoplanin plus ciprofloxacin on FN episodes and bacterial BSIs was also manifested among HD-cytarabine courses.

According to our study, teicoplanin seems an attractive agent with Gram-positive coverage to add to ciprofloxacin. Boztug, et al. [13] administered teicoplanin three days per week at a higher dose than usual, which appeared to be safe. Compared to daily dosing, such an alternate dosing schedule not only improves the quality of life but also minimizes the antibiotic exposure and handling of central venous lines. These advantages are of major interest as a decreased susceptibility to vancomycin and teicoplanin of *Enterococcus* species [23] and an increase in VRE strains in rectal cultures [15] have been observed with the use of vancomycin prophylaxis. One study also observed VRE BSIs with the use of vancomycin prophylaxis [15], whereas others did not [14,23]. Some patients in the study of Boztug, et al. [13] received prophylactic vancomycin monotherapy, but no VRE BSIs occurred and there was no increase in VRE in surveillance stool cultures. In our study, VRE BSIs were not observed. Unfortunately, we could not evaluate whether there was an increase in VRE in rectal surveillance cultures with the use of teicoplanin prophylaxis, as these data were not available for VUmc patients.

Besides the randomized trial that compared levofloxacin prophylaxis with no prophylaxis [24], there have been no other recent randomized trials published on the effect of antibacterial prophylaxis regimens in pediatric AML. The unsatisfactory results with the use of levofloxacin prophylaxis in terms of the GNR and VGS BSI incidences and the promising results with teicoplanin prophylaxis, administered on alternate days as well, support the conduction of a randomized trial including teicoplanin prophylaxis. Currently, the Pro-Teico study (EudraCT number 2020-000508-13), which is an international, multicenter, open-label, randomized clinical trial among pediatric AML patients, is ongoing within the NOPHO-DB-SHIP (Nordic countries, the Netherlands, Belgium, Spain, Hong Kong, Israel, and Portugal) consortium to compare the efficacy of intravenous teicoplanin prophylaxis three times per week with no prophylaxis, while patients are stratified for the administration of concomitant fluoroquinolone prophylaxis. In this trial, possible disadvantages, such as the emergence of VRE, are also being addressed, together with the apparent advantages.

This is the first study among Dutch pediatric AML patients and one of few studies reporting on the effect of antibacterial prophylaxis regimens in these patients. Since June 2018, Dutch pediatric oncology care has been centralized in the Princess Máxima Center (Utrecht, The Netherlands). Therefore, this study serves as a baseline measurement for future comparisons in our center. Our study is limited by its retrospective and non-randomized nature. Additionally, several risk factors for the occurrence of bacterial BSIs have not been studied like the presence of severe oral mucositis, previous bacterial BSIs, non-adherence to antimicrobial prophylaxis that had to be taken orally at home, change in and duration of empiric systemic antibiotics, or change in prophylaxis regimen after (repeated) infections. Nonetheless, we consider our study as highly informative, reflecting everyday clinical practice, and supporting the use of antibacterial prophylaxis in pediatric AML. If a decrease in the incidence of FN episodes and bacterial BSIs throughout AML treatment can be achieved, this may not only result in decreased infection-related morbidity and ultimately mortality, as demonstrated here in terms of fewer bacterial BSI-related PICU admissions, and fewer treatment delays, but also in a better quality of life for both parents and children, and lower health care costs. However, the benefits and drawbacks of the routine use of antibacterial prophylaxis need to be weighed continuously and may be institution-specific.

## 5. Conclusions

To conclude, in our study among Dutch pediatric AML patients, prophylaxis with teicoplanin plus ciprofloxacin resulted in a significant decrease in FN episodes and bacterial BSIs and in no bacterial BSI-related PICU admissions or deaths. This prophylactic regimen seems feasible and highly effective and needs validation in future studies.

## Figures and Tables

**Table 1 cancers-14-03172-t001:** Clinical characteristics of pediatric acute myeloid leukemia patients.

Characteristic	*N* = 82
Study period, *n* (%)	
1998–2018	41 (50)
(VUmc)	
2018–2021	41 (50)
(Princess Máxima Center)	
Sex, *n* (%)	
Male	45 (54.9)
Female	37 (45.1)
Age at diagnosis (years)	
Median (IQR)	9.0 (3.0–13.0)
WBC (×10^9^/L)	
Median (IQR)	14.3 (5.4–49.6)
FAB-type, *n* (%)	
M0	4 (4.9)
M1	8 (9.8)
M2	4 (4.9)
M4	21 (25.6)
M5	13 (15.9)
M6	2 (2.4)
M7	6 (7.3)
Unknown	24 (29.3)
CNS involvement, *n* (%)	
Negative	63 (76.8)
Positive	14 (17.1)
Unknown	5 (6.1)
Treatment protocol, *n* (%)	
ANLL-97	10 (12.2) *
AML-15	8 (9.8) †
DB-AML-01	11 (13.4)
NOPHO-DBH AML 2012	53 (64.6)

* Two patients started with ANLL-97, but received consolidation therapy following AML-15. ^†^ One patient started with AML-15, but received consolidation therapy following DB-AML-01. Abbreviations: CNS, central nervous system; FAB, French-American-British; IQR, interquartile range; *N*, number of patients; VUmc, VU University Medical Center (Amsterdam, The Netherlands); WBC, white blood cell count.

**Table 2 cancers-14-03172-t002:** The incidence of febrile neutropenic episodes, bacterial bloodstream infections, and bacterial bloodstream infection-related pediatric intensive care unit admissions according to antibacterial prophylaxis regimen in pediatric acute myeloid leukemia patients.

Antibacterial Prophylaxis Regimen *	*N* of Courses	FN Episode		*P*	Bacterial BSI		*P*	Gram-positive BSI		*P*	VGS BSI		*P*	Gram-negative BSI		*P*	PICU		*P*
		*N*	%		*N*	%		*N*	%		*N*	%		*N*	%		*N*	%	
Ciprofloxacin	92	83	90.2	**<0.0001 ^†^**	58	63.0	**<0.0001 ^†^**	53	57.6	**<0.0001 ^†^**	22	23.9	**0.0005 ^†^**	7	7.6	**0.004 ^†^**	7	7.6	**0.029 ^†^**
Penicillin plus ciprofloxacin	138	104	75.4		46	33.3		45	32.6		19	13.8		1	0.7		3	2.2	
Teicoplanin plus ciprofloxacin	51	22	43.1		2	3.9		2	3.9		0	0		0	0		0	0	
Ciprofloxacin	92	83	90.2	**0.005 ^‡^**	58	63.0	**<0.0001 ^‡^**	53	57.6	**0.0002 ^‡^**	22	23.9	**0.055 ^‡^**	7	7.6	**0.008 ^‡^**	7	7.6	0.094 ^‡^
Penicillin plus ciprofloxacin	138	104	75.4		46	33.3		45	32.6		19	13.8		1	0.7		3	2.2	
Penicillin plus ciprofloxacin	138	104	75.4	**<0.0001 ^‡^**	46	33.3	**<0.0001 ^‡^**	45	32.6	**<0.0001 ^‡^**	19	13.8	**0.002 ^‡^**	1	0.7	1.000 ^‡^	3	2.2	0.565 ^‡^
Teicoplanin plus ciprofloxacin	51	22	43.1		2	3.9		2	3.9		0	0		0	0		0	0	
No prophylaxis	10	9	90	NA	6	60	NA	4	40	NA	3	30	NA	3	30	NA	1	10	NA
Penicillin	12	12	100	NA	6	50	NA	2	16.7	NA	2	16.7	NA	4	33.3	NA	3	25	NA
Macrolide plus ciprofloxacin	13	10	76.9	NA	7	53.8	NA	7	53.8	NA	2	15.4	NA	0	0	NA	1	7.7	NA

* The antibacterial prophylaxis regimen switched during the neutropenic period of 13 evaluable courses, namely from initially no prophylaxis to single-agent penicillin (*n* = 1), single-agent ciprofloxacin to penicillin plus ciprofloxacin (*n* = 4), single-agent ciprofloxacin to a macrolide plus ciprofloxacin (*n* = 1), penicillin plus ciprofloxacin to teicoplanin plus ciprofloxacin (*n* = 3), penicillin plus ciprofloxacin to single-agent ciprofloxacin (*n* = 2), and penicillin plus ciprofloxacin to a macrolide plus ciprofloxacin (*n* = 2). **^†^** Pearson’s χ^2^ test. ^‡^ Fisher’s exact test. NOTE. Bold states statistical significance. Due to polymicrobial BSIs, the number of bacterial BSIs may not correspond with the accumulated number of Gram-positive and Gram-negative BSIs. Because of the small number of courses with no prophylaxis (*n* = 10), single-agent penicillin (*n* = 12), and a macrolide plus ciprofloxacin (*n* = 13), these regimens were not statistically compared. Abbreviations: BSI, bloodstream infection; FN, febrile neutropenic; N, number; NA, not applicable; PICU, pediatric intensive care unit; VGS, viridans group streptococci.

**Table 3 cancers-14-03172-t003:** Microorganisms detected in blood cultures of febrile neutropenic pediatric acute myeloid leukemia patients.

Microorganism	Total		Ciprofloxacin		Penicillin plus Ciprofloxacin		Teicoplanin plus Ciprofloxacin		No Prophylaxis		Penicillin		Macrolide plus Ciprofloxacin	
	*N*	%	*N*	%	*N*	%	*N*	%	*N*	%	*N*	%	*N*	%
Gram-positive bacteria	138	85.7	70	84.3	52	98.1	2	66.7	4	57.1	2	28.6	8	100
Coagulase-negative staphylococci	71	44.1	36	43.4	26	49.1	2	66.7	1	14.3	0	0	6	75
Viridans streptococci	48	29.8	22	26.5	19	35.8	0	0	3	42.9	2	28.6	2	25
*Micrococcus* spp.	3	1.9	3	3.6	0	0	0	0	0	0	0	0	0	0
*Rothia mucilaginosa*	8	5.0	5	6.0	3	5.7	0	0	0	0	0	0	0	0
Other *	8	5.0	4	4.8	4	7.5	0	0	0	0	0	0	0	0
Gram-negative bacteria	16	9.9	7	8.4	1	1.9	0	0	3	42.9	5	71.4	0	0
*Escherichia coli*	7	4.3	2	2.4	0	0	0	0	2	28.6	3	42.9	0	0
Other ^†^	9	5.6	5	6.0	1	1.9	0	0	1	14.3	2	28.6	0	0
Fungi/Yeast ^‡^	7	4.3	6	7.2	0	0	1	33.3	0	0	0	0	0	0
Total	161	100	83	100	53	100	3	100	7	100	7	100	8	100

* Two *Staphylococcus aureus*, one *Enterococcus faecium*, one *Bacillus cereus*, one *Granulicatella adiacens*, one *Bacillus megaterium*, one *Sporosarcina* sp., one *Brevibacterium* sp. ^†^ Two *Pseudomonas aeruginosa*, one *Enterobacter cancerogenus*, one *Roseomonas* sp., one *Acinetobacter baumanii*, one *Pseudomonas putida*, one *Klebsiella pneumoniae*, one *Fusobacterium* sp., one *Stenotrophomonas maltophilia*. ^‡^ Two *Candida non-albicans* not further specified, two *Candida krusei*, one *Candida glabrata*, one *Saprochaete clavata*, one *Fusarium* sp. NOTE. Polymicrobial blood cultures are included.

**Table 4 cancers-14-03172-t004:** The incidence of febrile neutropenic episodes, bacterial bloodstream infections, and bacterial bloodstream infection-related pediatric intensive care unit admissions according to low-dose and high-dose cytarabine courses and per antibacterial prophylaxis regimen.

Group *	*N* of Courses	FN Episode		*P*	Bacterial BSI		*P*	Gram-positive BSI		*P*	VGS BSI		*P*	Gram-negative BSI		*P*	PICU		*P*
		*N*	%		*N*	%		*N*	%		*N*	%		*N*	%		*N*	%	
LD-cytarabine	19	12	63.2	0.149 ^‡^	3	15.8	**0.028 ^‡^**	3	15.8	0.051 ^‡^	0	0	0.052 ^‡^	0	0	0.610 ^‡^	0	0	0.610 ^‡^
HD-cytarabine	284	224	78.9		122	43.0		110	38.7		48	16.9		15	5.3		15	5.3	
LD-cytarabine																		
Ciprofloxacin	4	4	100	0.075 ^†^	1	25.0	0.338 ^†^	1	25.0	0.338 ^†^	0	0	NA	0	0	NA	0	0	NA
Penicillin plus ciprofloxacin	9	3	33.3		2	22.2		2	22.2		0	0		0	0		0	0	
Teicoplanin plus ciprofloxacin	8	5	62.5		0	0		0	0		0	0		0	0		0	0	
No prophylaxis	0	-	-		-	-		-	-		-	-		-	-		-	-	
Penicillin	0	-	-		-	-		-	-		-	-		-	-		-	-	
Macrolide plus ciprofloxacin	0	-	-		-	-		-	-		-	-		-	-		-	-	
HD-cytarabine																		
Ciprofloxacin	88	79	89.8	**<0.0001 ^†^**	57	64.8	**<0.0001 ^†^**	52	59.1	**<0.0001 ^†^**	22	25	**0.001 ^†^**	7	8.0	**0.005 ^†^**	7	8.0	**0.038 ^†^**
Penicillin plus ciprofloxacin	129	101	78.3		44	34.1		43	33.3		19	14.7		1	0.8		3	2.3	
Teicoplanin plus ciprofloxacin	43	17	39.5		2	4.7		2	4.7		0	0		0	0		0	0	
No prophylaxis	10	9	90	NA	6	60	NA	4	40	NA	3	30	NA	3	30	NA	1	10	NA
Penicillin	12	12	100	NA	6	50	NA	2	16.7	NA	2	16.7	NA	4	33.3	NA	3	25	NA
Macrolide plus ciprofloxacin	13	10	76.9	NA	7	53.8	NA	7	53.8	NA	2	15.4	NA	0	0	NA	1	7.7	NA

***** The analyses comparing the LD- and HD-cytarabine courses are based on the 303 evaluable courses. The analyses comparing the antibacterial prophylaxis regimens among LD- and HD-cytarabine courses are based on 316 courses, as the antibacterial prophylaxis regimen switched during the neutropenic period of 13 courses. ^†^ Pearson’s χ^2^ test. ^‡^ Fisher’s exact test. NOTE. Bold states statistical significance. HD-cytarabine was defined as ≥1000 mg/m^2^ cytarabine per course. Because of the small number of courses with no prophylaxis (*n* = 10), single-agent penicillin (*n* = 12), and a macrolide plus ciprofloxacin (*n* = 13), these regimens were not statistically compared. Abbreviations: BSI, bloodstream infection; FN, febrile neutropenic; HD, high-dose; LD, low-dose; N, number; NA, not applicable; PICU, pediatric intensive care unit; VGS, viridans group streptococci.

**Table 5 cancers-14-03172-t005:** Penicillin susceptibility of viridans group streptococci isolates and per antibacterial prophylaxis regimen.

Group	Penicillin Susceptibility Known/ Total VGS Isolates	Penicillin S	Penicillin I	Penicillin R
Total	39/48	20	7	12
No prophylaxis	3/3	1	1	1
Ciprofloxacin	18/22	12	1	5
Penicillin	2/2	1	1	0
Penicillin plus ciprofloxacin	14/19	5	3 *	6
Macrolide plus ciprofloxacin	2/2	1	1	0

* One VGS isolate was penicillin I/R. Abbreviations: I, intermediate-sensitive; R, resistant; S, sensitive; VGS, viridans group streptococci.

## Data Availability

The data presented in this study may be available from the corresponding author upon reasonable request.

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
