# Peer review of "Effect of Antibacterial Prophylaxis on Febrile Neutropenic Episodes and Bacterial Bloodstream Infections in Dutch Pediatric Patients with Acute Myeloid Leukemia: A Two-Center Retrospective Study"

_cancers, 2022, doi:10.3390/cancers14133172_

Round 1

Reviewer 1 Report

Van Weelderen et al report a retrospective study on different antibiotic regimens for prophylaxis of bacterial infections following chemotherapy for acute myeloid leukemia in pediatric patients. The paper is well written and the different antibiotic regimens with corresponding infectious results are adequately compared. A regimen with teicoplanin and ciprofloxacin yields superior results regarding episodes of febrile neutropenia, blood stream infections and PICU admissions. Regarding usage of ciprofloxacin, the authors should mention that this medication is only approved in Europe for treatment of severe infections in pediatric patients and hence administration as prophylaxis is off-label. Possible side effects such as arthropathies with longer administration should be considered. Results are obviously limited by the retrospective study design without randomization of the different regimens, hence the authors refer to the ongoing randomized NOPHO study. In conclusion, this study adds valuable data to the issue of antibiotic prophylaxis in pediatric AML and is recommended for publication.

Author Response

Response to Reviewer 1 Comments

Point 1: Regarding usage of ciprofloxacin, the authors should mention that this medication is only approved in Europe for treatment of severe infections in pediatric patients and hence administration as prophylaxis is off-label.

Response 1: We agree with the reviewer that this is important to the readers. We added in the method section (2.4) that ciprofloxacin as well as teicoplanin and macrolides do not have an approved indication to prevent bloodstream infections in children in Europe and that the administration of these antibiotics as prophylaxis was off-label.

Point 2: Possible side effects such as arthropathies with longer administration should be considered.

Response 2: Fortunately, no musculoskeletal toxicities were reported in our study population. We added this to the results section and we discussed this potential rare side effect in the discussion section (covered by the term musculoskeletal toxcities).

Reviewer 2 Report

The article retrospectively evaluates the effect of prophylactic antibacterial regimens on the occurrence of fevers and bacterial BSIs on a sample of eighty-two pediatric AML patients at two Dutch medical centers from May 1998 to June 2018. When applied to this population, the study found that prophylactic teicoplanin and ciprofloxacin significantly reduced febrile neutropenic episodes, bacterial blood stream infections, and bacterial BSI-related PICU admission or deaths compared to other regimens. These findings have important implications for both clinicians and researchers.

Strengths

·         Introduction. Authors clearly explains the motivation of the study by providing context to appreciate AML treatment complications, need for prophylactic interventions, and potential impact of the study results.

·         Discussion. The authors provide relevant results and describe limitations from similar studies.

·         Discussion. The authors illustrate the impact of their study on the field, while acknowledging to its limitations.

·         Discussion. The authors provide description of a relevant current international study.

Weaknesses

·         Discussion. Authors did not discuss the location characteristics of antimicrobial prophylaxis administration (hospital vs outpatient clinic vs domiciliary care). The authors may wish to include this as an additional limitation.

·         Supplementary Materials. Authors provided a broken web link (www.mpdi.com/xx/s1) for supporting information. Therefore, I was unable to appreciate the treatment protocols and results that were excluded from analysis.

·         Results. Table 2-4  were difficult to interpret. Authors may want to revise formatting.

·         Writing. Found writing to be unclear and ineffective at times due to many poorly structured sentences. Examples provided below.

Revisions to paper:

·         Line 57: Delete “Especially”

·         Line 76: Change “taking” to “taken”

·         Line 266-267: Change “Among HD-cytarabine courses, teicoplanin plus ciprofloxacin was significantly associated  with less FN episodes..” to “Among HD-cytarabine courses, teicoplanin plus ciprofloxacin was associated with significantly less FN episodes..”

·         Line 327-328: Change “Three BSI-related deaths were caused by Gram-negative microorganisms, all prior to the introduction of ciprofloxacin prophylaxis.” To “Prior to the introduction of ciprofloxacin prophylaxis, there were three BSI-related deaths caused by Gram-negative microorganisms.”

·         Line 330: Delete “an”

·         Line 425: Change “Compared with daily dosing” to “Compared to daily dosing”

·         Line 426-427: Change “but also minimizes the antibiotic exposure and central venous lines are less often handled” to “but also minimizes the antibiotic exposure and handling of central venous lines”

·         Line 470: Change “delay’s to “delays”

Author Response

Response to Reviewer 2 Comments

Point 1: Discussion. Authors did not discuss the location characteristics of antimicrobial prophylaxis administration (hospital vs outpatient clinic vs domiciliary care). The authors may wish to include this as an additional limitation.

Response 1: Teicoplanin was administered intravenously, either at the hospital, outpatient clinic, or via domiciliary care. Therefore, we know adherence to teicoplanin was high. With respect to the other prophylactic agents, which were mainly given orally and taken at home (as Dutch patients will be discharged home after the chemotherapy course if their condition allows this), we acknowledge that we can’t be completely certain how the adherence to these agents was. However, adherence to therapy is in general relatively good in the Netherlands. The limitations paragraph in the discussion described ‘’non-adherence to therapy’’ as one of the limitations. We have now changed this into ‘’non-adherence to antimicrobial prophylaxis that had to be taken orally at home’’. 

Point 2: Supplementary Materials. Authors provided a broken web link (www.mpdi.com/xx/s1) for supporting information. Therefore, I was unable to appreciate the treatment protocols and results that were excluded from analysis.

Response 2: We are sorry to hear that you were unable to appreciate the treatment protocols and results that were excluded from analysis. The corresponding coauthor contacted the editor and was told that the functional website link you are referring to will be available when the manuscript is published. The supporting information was also uploaded as a Word file and available for reviewers to download from the system.

Point 3: Results. Table 2-4  were difficult to interpret. Authors may want to revise formatting.

Response 3: The corresponding author asked the editor how the Tables were shown for the reviewers and was told that the Tables were fitted according to the MDPI layout style. This style is different from how the Tables were submitted. We agree that the MDPI layout style is difficult to interpret as Tables 2-4 really need to be shown in landscape orientation. In the revised version of the manuscript, Tables 2-4 are re-fitted into landscape orientation. Hopefully, these Tables are now easier to interpret.

Point 4: Writing. Found writing to be unclear and ineffective at times due to many poorly structured sentences. Examples provided below.

Response 4: We have processed the suggested revisions to the paper. Additionally, we have asked a native English speaker to proof-read the manuscript in order to increase the readability.

Revisions to paper:
·         Line 57: Delete “Especially”
Adjusted accordingly.

  • Line 76: Change “taking” to “taken”
    Adjusted accordingly.
  • Line 266-267: Change “Among HD-cytarabine courses, teicoplanin plus ciprofloxacin was significantly associated  with less FN episodes..” to “Among HD-cytarabine courses, teicoplanin plus ciprofloxacin was associated with significantly less FN episodes..”
    The order of the sentence has been changed.
  • Line 327-328: Change “Three BSI-related deaths were caused by Gram-negative microorganisms, all prior to the introduction of ciprofloxacin prophylaxis.” To “Prior to the introduction of ciprofloxacin prophylaxis, there were three BSI-related deaths caused by Gram-negative microorganisms.”
    The sentence has been changed.
  • Line 330: Delete “an”
    Adjusted accordingly.
  • Line 425: Change “Compared with daily dosing” to “Compared to daily dosing”
    Adjusted accordingly.
  • Line 426-427: Change “but also minimizes the antibiotic exposure and central venous lines are less often handled” to “but also minimizes the antibiotic exposure and handling of central venous lines”
    The sentence has been changed.
  • Line 470: Change “delay’s to “delays”
    Adjusted accordingly.
